# Synthesis and Frictional Characteristics of Bio-Based Lubricants Obtained from Fatty Acids of Castor Oil

Paulo Roberto Campos Flexa Ribeiro Filho [1,2], Matheus Rocha do Nascimento [1], Silvia Shelly Otaviano da Silva [1], Francisco Murilo Tavares de Luna [1] 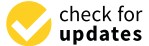, Enrique Rodríguez-Castellón [3,*] and Célio Loureiro Cavalcante, Jr. [1,*]

1  Grupo de Pesquisa em Separações por Adsorção (GPSA), Departamento de Engenharia Química, Campus do Pici, Universidade Federal do Ceará, Fortaleza 60455760, Brazil
2  Departamento de Engenharia Mecânica, Universidade Estadual do Maranhão, São Luís 65055310, Brazil
3  Departamento de Química Inorgánica, Cristalografía y Mineralogía, Facultad de Ciencias, Campus Teatino, Universidad de Málaga, 29071 Málaga, Spain
*  Correspondence: castellon@uma.es (E.R.-C.); celio@gpsa.ufc.br (C.L.C.J.); Tel.: +34-645263909 (E.R.-C.); +55-85-3366-9611 (C.L.C.J.)

**Abstract:** The depletion of oil reserves and concerns about the environmental impact of the use and incorrect disposal of mineral lubricants have been promoting the development of bio-based lubricants. In this study, biolubricants obtained from fatty acids of castor oil were synthesized by esterification (>wt.%93), epoxidation (>wt.%92), and oxirane ring opening reactions using water (>wt.%92) or 2-ethylhexanol (>wt.%94) as nucleophilic agents. The frictional characteristics of the synthesized samples were obtained through tribological tests performed in a four-ball tester and compared with a commercial mineral oil. The sample obtained through oxirane ring opening with water showed the best frictional performance (FC = 0.0699 ± 0.0007) among the prepared samples, with equivalent wear rate (WSD = 281.2 ± 5.54 µm) and ca. 20% lower friction coefficient when compared to the commercial mineral oil, indicating its great potential for replacing mineral fossil oils.

**Keywords:** tribology; biolubricant; oxirane ring opening

## 1. Introduction

Lubrication, which is the application of a fluid film between the contacting metal surfaces in static and dynamic operating conditions, is used to minimize the effects of friction and wear on machinery and equipment. Currently, most lubricants are produced from mineral oil; although these lubricants meet technical requirements, their production is limited by the availability of oil reserves. Increased worldwide concerns about the environmental impact of the incorrect disposal of mineral oils and emissions of metals generated during the operation of internal combustion engines led, for example, to the introduction of non-technical criteria (ecotoxicological and sustainability properties) to reduce the environmental impact of lubricants [1–4].

Thus, efforts are devoted to the development of bio-based lubricants produced from renewable and biodegradable sources. Vegetable oils, particularly non-edible vegetable oils, are used as the main basis for the production of biolubricants. Although edible vegetable oils may serve as a basis for the production of biolubricants, such usage is associated with sustainability concerns because increased demand for edible oils would result in the expansion of plantations which could lead to deforestation and eutrophication [1,5]. At the same time, non-edible oilseeds can be cultivated in low-fertility dry and semi-arid environments [6].

Castor oil stands out among non-edible vegetable oils because of its main composition of hydroxylated fatty acids [4]. Ricinoleic acid comprises wt.% 82–90 of fatty acids in castor oil (*Ricinus communis L.*) [7–9]. Ricinoleic acid contains a hydroxyl group (–OH) in its chain, which provides good lubricity, especially for applications as hydraulic fluid, in

which the lubricant operates in the mixed lubrication/EHL and hydrodynamic regimes since the hydroxyl group (-OH) forms a thick and viscous lubricant due to intermolecular hydrogen bonding. [10–13]. Although ricinoleic acid has good tribological properties, the presence of a double bond in its structure increases its susceptibility to oxidation, which is disadvantageous for direct applications in mechanical systems [14,15].

To improve the oxidative stability and other physicochemical properties of unsaturated vegetable oils, studies have been carried out to obtain esters through chemical processes [16,17]. The most common chemical modification routes are transesterification/esterification, hydrogenation, estolide formation, and epoxidation reactions [18–20]. The transesterification route consists of replacing the glycerol portions of the triglyceride molecule of vegetable oils with a long- or branched-chain alcohol and performing esterification with free fatty acids extracted from vegetable oils [19,21]. Hydrogenation consists of the isomerization of cis- and trans-acids. Estolide esters are obtained by bonding a carboxylic acid group to the double bond of another fatty acid [22]. Epoxidation is the removal of double carbon bonds through the introduction of an oxygen atom, resulting in the formation of an epoxide functional group [20]. Although epoxidation of vegetable oils removes unsaturations, it yields biolubricants with poor low-temperature properties, which need to be further submitted to an oxirane ring opening reaction and subsequent esterification [16–22]. The possibility of using a variety of vegetable oils, alcohols, and other materials results in a wide variety of products with diverse physicochemical and tribological characteristics.

Recent studies have used castor oil and/or ricinoleic acid to obtain biolubricants via an estolide formation route [23]. The obtained products showed a similar pour point and higher oxidative stability than castor oil-based lubricants reported in the literature. Encinar et al., (2020) used castor oil transesterification to obtain biolubricants with a higher flash point than mineral oils [8]. Saboya et al., (2017) obtained castor oil-based biolubricants by esterification and demonstrated that 2-ethylhexyl ricinoleate exhibits excellent low-flow properties, viscosity index (VI) comparable to that of commercial synthetic oils, biodegradability higher than mineral oil, and better oxidative stability than castor oil [24]. Rios et al., (2020) prepared biolubricants through epoxidation followed by an oxirane ring opening, and the obtained products exhibited excellent low-temperature properties with high oxidative stability [7].

Although the mentioned studies report significant improvements in physicochemical properties, neither of them evaluated the frictional characteristics of the obtained products. Tribological tests simulate lubricant operation, and proper performance (high energy efficiency, low friction coefficients (FC), and wear scar diameter (WSD)) in the test is paramount to validate the use of the considered lubricant in mechanical systems [13,25–27]. The present study aimed at producing biolubricants from fatty acids of castor oil through esterification, epoxidation, and oxirane ring opening reactions with 2-ethylhexanol and water used as nucleophilic agents to produce the ether branches, since these consecutive reactions improve their properties at low temperatures, viscosity index, and oxidative stability [13]. The study of their tribological characteristics becomes of fundamental importance to meet all the technical criteria for their application in machinery lubrication. Therefore, the main physicochemical properties of the products were determined, and the tribological characteristics were evaluated using a four-ball tester.

## 2. Materials and Methods

### 2.1. Materials

A sample of fatty acids from castor oil was purchased from Azevedo Ind. e Com. de Óleos LTDA (São Paulo, Brazil). The sample presented in its composition a mixture of ricinoleic acid (82–90 wt.%), linoleic acid (wt.% 2–8), oleic acid (wt.% 2–7), stearic acid (2 wt.%), and palmitic acid (wt.% 2). Acetone (>wt.% 99.5), p-toluenesulfonic acid (>wt.% 98), toluene (>wt.% 99), and sodium bicarbonate (>wt.% 99) were supplied by Neon (Suzano, Brazil). Hydrogen peroxide (35.5% *v/v*), anhydrous sodium sulfate (>wt.% 99),

and formic acid (>wt.% 85) were purchased from Dinâmica (São Paulo, Brazil). Amberlyst 15 resin, 2-ethylhexanol (>wt.% 99), potassium bromide, and deuterated chloroform (CDCl3, 99.8%) were supplied by Sigma-Aldrich (San Luis, Missouri, USA). A commercial mineral oil sample (SAE 20W50) was purchased from Maxon Oil (São José dos Pinhas, Brazil), with physicochemical properties listed in Table 1.

**Table 1.** Physicochemical properties of the commercial mineral oil (SAE 20W50).

| Property | Value | Method |
|---|---|---|
| Density at 20 °C (g/cm$^3$) | 0.87 | ASTM D7042/NBR 7148 |
| Kinematic viscosity at 40 °C (cSt) | 152.1 | ASTM D445/NBR 10441 |
| Kinematic viscosity at 100 °C (cSt) | 18.04 | ASTM D445/NBR 10441 |
| Viscosity index | 132 | ASTM D2270/NBR 14358 |
| Open cup flash point (°C) | 223 | ASTM D921/NBR 11341 |

### 2.2. Synthesis Procedure

Biolubricants were obtained using the synthesis route adapted from the previous studies [7,28]. A scheme of the reactions used to synthesize the bio-based lubricants is shown in Figure 1. The sample of fatty acids from castor oil (FACO) was submitted to esterification (Figure 1b), epoxidation (Figure 1c), and oxirane ring opening reactions (Figure 1d,e).

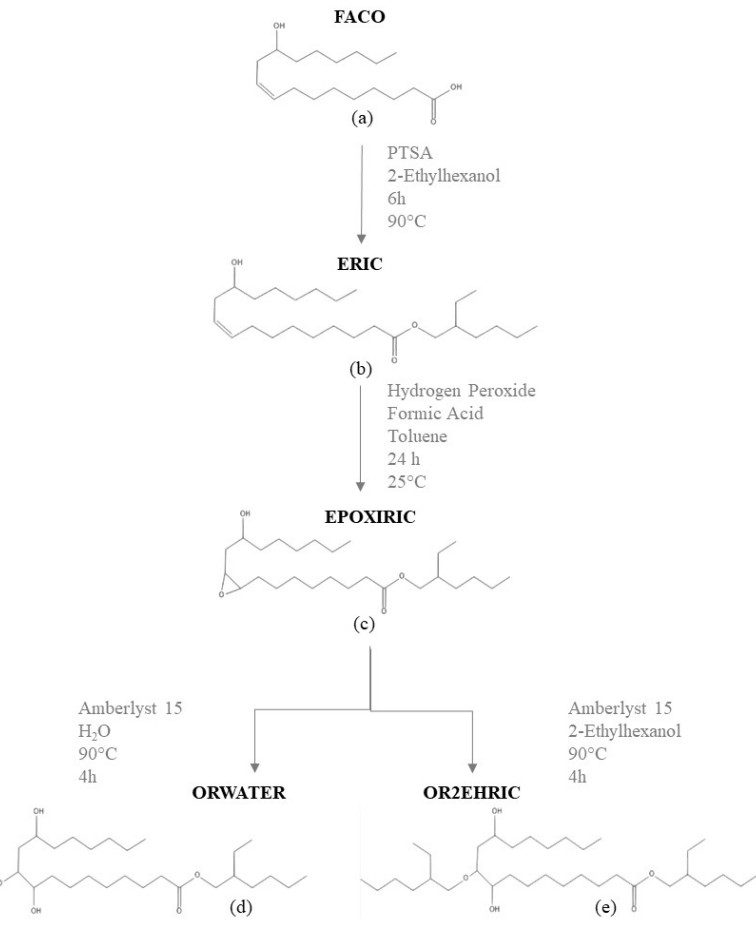

**Figure 1.** Scheme of the synthesis routes used to obtain bio-based lubricants. (a) FACO (fatty acids from castor oil), (b) ERIC (esterification reaction product), (c) EPOXIRIC (epoxidation reaction product), (d) ORWATER (product of the oxirane ring opening reaction with water as a nucleophilic agent), and (e) OR2EHRIC (product of the oxirane ring opening reaction with 2-ethylhexanol as a nucleophilic agent).

Esterification was performed in a 3-neck flask, where 100 g (0.336 mol) of fatty acids from castor oil (FACO) were mixed with 131 g (1.01 mol) of 2-ethylhexanol (molar ratio FACO:2-ethylhexanol = 1:3) and 10 g of p-toluenesulfonic acid (10% of fatty acids by weight). The reaction was conducted for 6 h at 90 °C, under reflux and constant stirring at 900 rpm. Then, the mixture was transferred to a separatory funnel, and the organic phase was washed with a solution of $NaHCO_3$ (5% *w/v*) and distilled water until pH 7 was attained. The product was dried with anhydrous $Na_2SO_4$ and distilled in a Kugelrohr system under vacuum ($3 \cdot 10^{-2}$ mbar) at 110 °C to remove the excess of 2-ethylhexanol. The final product of this step was named ERIC.

Next, the ERIC sample was submitted to epoxidation: in a flat-bottom flask, a solution of 70 g (0.171 mol) of ERIC, 7.4 mL (0.171 mol) of formic acid, and 50 mL of toluene were prepared. Subsequently, 57 mL (0.684 mol) of the hydrogen peroxide solution (molar ratio ERIC:$CH_2O_2$:$H_2O_2$ 1:1:4) was added dropwise to the reaction solution. The reaction was conducted for 24 h at room temperature and under constant stirring at 900 rpm. Then, the liquid was transferred from the flask to a separatory funnel. The upper phase was neutralized with a solution of $NaHCO_3$ (5% *w/v*), washed with distilled water, and dried with anhydrous sodium sulfate. The toluene was removed using a rotary evaporator, under reduced pressure at 90 °C for 40 min. The product of this step was named EPOXIRIC.

Via the oxirane ring opening reaction, two products were obtained from the EPOXIRIC sample with 2-ethylhexanol (named OR2EHRIC) and with water (named ORWATER) used for the nucleophilic attack with molar ratios of 10:1 water/EPOXIRIC and 3:1 alcohol/EPOXIRIC. Dry Amberlyst 15 (A15) was used as a catalyst at a mass ratio (EPOXIRIC/A15) of 10:1. The reaction was conducted in a three-neck flask at 90 °C for 4 h under reflux and constant magnetic stirring (900 rpm). Subsequently, the product was filtered to separate the catalyst and washed in a separatory funnel with a solution of sodium bicarbonate (5% *w/v*) and distilled water, until reaching pH 7. The samples were dried with anhydrous $Na_2SO_4$. Then, the ORWATER (ring opening with water) sample was stored, and the OR2EHRIC (ring opening with 2-ethylhexanol) product was distilled in a Kugelrohr system under vacuum ($3 \cdot 10^{-2}$ mbar) at 110 °C to remove excess 2-ethylhexanol.

### 2.3. Physicochemical Characterization

All samples were analyzed by one-dimensional proton nuclear magnetic resonance spectroscopy ($^1$H NMR; Bruker Avance DRX-500, Billerica, Massachusetts, USA) operating at 500 MHz with deuterated chloroform as a solvent.

Fourier-transform infrared spectroscopy (FTIR; Shimadzu IRTracer-100, Quioto, Japan) was performed for all samples to confirm the esterification, epoxidation, and oxirane ring opening reactions. The analysis was carried out using a potassium bromide (KBr) tablet in the scanning range of 400–4000 cm$^{-1}$ [29]. The tablet was prepared by pressing at a force of 8 kN. Thirty-two scans were performed at a resolution of 4 cm$^{-1}$.

Density at 20 °C and kinematic viscosity at 40 °C and 100 °C of all samples were determined following ASTM D7042 and ASTM D445 methods, respectively, in an Anton Paar SVM 3000 equipment (Graz, Austria) [30,31]. The ASTM D2270 method was used to obtain the viscosity index value (VI) [32].

### 2.4. Tribological Evaluation

All samples were taken to a tribological test, using a four-ball tester coupled to a DHR-3 rheometer (TA Instruments, New Castle, Delaware, USA). More information about the tribological tester may be found in reference [33]. During the test, one ball rotated at a constant sliding speed (0.228 m/s), temperature of 40 °C, and under loading force (55 N) against three fixed balls submerged in the evaluated lubricant, as shown in Figure 2. The test was conducted for 1 h. The balls used in this test were made of chrome steel alloy (AISI52100) with a hardness of 64 HRC, diameter of 12.7 mm, and an initial surface roughness of 0.015 μm. Prior to the test, the balls were cleaned with acetone and dried under ambient conditions. Optical microscopy (Zeiss, Oberkochen, Germany) was used

to evaluate the wear morphology and determine the WSD for each test. New balls were used for each new test. The mineral oil sample (SAE 20W50) was used as a reference for tribological comparison, using friction coefficients (FC) and wear scar diameters (WSD) of the biolubricants.

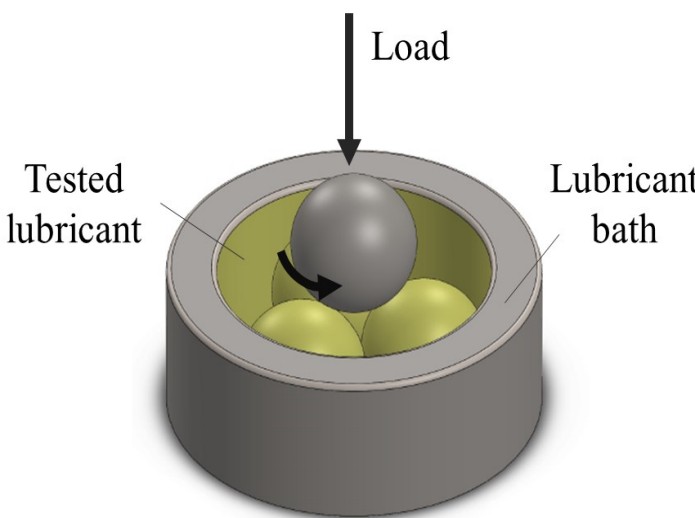

**Figure 2.** Four-ball tribological test setup.

### 3. Results and Discussion

*3.1. Physicochemical Characterization*

Densities at 20 °C, kinematic viscosities at 40 °C and 100 °C, and viscosity indices obtained for all samples are listed in Table 2. In lubricants, density is used for evaluation of used lubricants: an increase in density compared to the fresh lubricant may indicate the presence of insolubles, water, higher-density products, and oxidized compounds, while a decrease in density may indicate the presence of lower-density contaminants and/or fuel [34]. The ERIC, EPOXIRIC, and OR2EHRIC samples showed lower densities than the FACO sample, whereas the ORWATER sample presented a higher density than the FACO sample. These results may be related to the lower viscosities of the ERIC, EPOXIRIC, and OR2EHRIC samples and the higher viscosity of the ORWATER sample, if compared to the FACO sample, since density values depend on viscosity, quality, and additives content of the sample [35–37].

**Table 2.** Physicochemical properties of the FACO, ERIC, EPOXIRIC, OR2EHRIC, and ORWATER samples.

| Property | FACO | ERIC | EPOXIRIC | OR2EHRIC | ORWATER | Method |
|---|---|---|---|---|---|---|
| Density at 20 °C $(g/cm^3)$ | 0.941 | 0.896 | 0.935 | 0.928 | 0.960 | ASTM D7042 |
| Kinematic viscosity at 40 °C (cSt) | 137.02 | 25.20 | 46.68 | 67.33 | 420.46 | ASTM D445 |
| Kinematic viscosity at 100 °C (cSt) | 13.04 | 4.55 | 6.95 | 8.79 | 24.39 | |
| Viscosity index | 86.5 | 89.2 | 105.0 | 103.1 | 71.8 | ASTM D2270 |

Viscosity is a measurement of the shear strength of a lubricant and depends primarily on the molecular interactions within the lubricant [34]. The FACO sample is mainly composed of ricinoleic acid, which contains a hydroxyl group that enhances attractive intermolecular interactions due to hydrogen bonding. For this reason, its viscosity is higher than those of other fatty acids [7,38,39]. The lower viscosity of the ERIC sample if compared to the FACO sample may be ascribed to the introduction of a branched ethyl group that makes intermolecular interactions more difficult [7,40].

The insertion of the epoxy group into the EPOXIRIC sample led to an increase in viscosity because of the intermolecular interactions induced by the presence of an oxygen atom in the ring [41]. Through oxirane ring opening reactions, the OR2EHRIC and ORWATER samples were obtained using 2-ethylhexanol and water, respectively, as nucleophilic agents. The ORWATER sample showed higher viscosities due to the introduction of two new hydroxyl groups that strengthened the molecular interactions through hydrogen bonds [13]. At the same time, in the OR2EHRIC sample, an epoxy group was transformed into an alcohol group and an ether group. Due to the lower number of hydrogen bonds, the OR2EHRIC sample showed lower viscosities than ORWATER sample [7]. The viscosity index (VI) describes the effect of temperature on the viscosity of the lubricant. The higher the VI, the lower the influence of temperature on viscosity [42–44]. The studied samples presented VI values (71.8–105) within the limits of mineral lubricants (groups I and II), according to the API classification [45].

The FTIR spectra of all samples are shown in Figure 3. In the FACO sample, the band at 1710 cm$^{-1}$ was attributed to the carbonyl group (C=O) in the long-chain fatty acids. For the ERIC sample, this peak shifts from 1710 to 1740 cm$^{-1}$, indicating the presence of an ethyl ester C=O group. This result indicates that the functional groups of carboxylic acids present in the spectrum of the FACO sample were indeed transformed into esters after the esterification reaction [46]. Furthermore, for the ERIC sample, a peak may be observed at around 1170 cm$^{-1}$; this peak is absent in the FACO sample and is characteristic of the presence of the C=O functional group, from the ester formation [47,48]. In addition, the CH$_3$ group corresponding to the ethyl ester appears in the ERIC sample spectrum at 1461 cm$^{-1}$ [49].

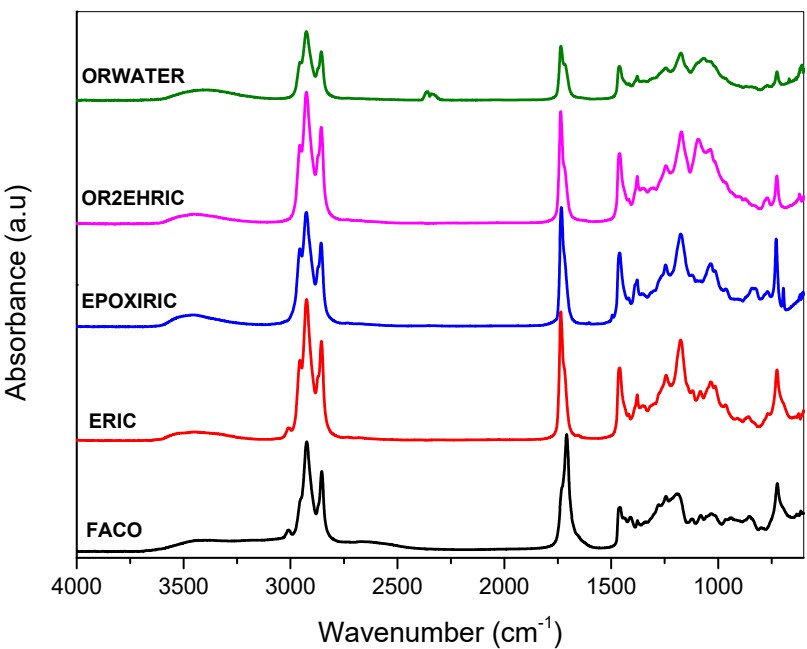

**Figure 3.** FTIR spectra of the FACO, ERIC, EPOXIRIC, OR2EHRIC, and ORWATER samples.

The differences between the spectra of the FACO and ERIC samples demonstrate the effective esterification of the fatty acids from castor oil. In the EPOXIRIC sample, bands at 825 and 841 cm$^{-1}$, ascribed to the epoxide groups, are observed. These bands are absent in the spectra of the OR2EHRIC and ORWATER samples, demonstrating the successful oxirane ring opening. In addition, in the OR2EHRIC and ORWATER samples, the bands at 1737 cm$^{-1}$ represent carbonyl elongation (C=O). Bands at 1090, 1172, and 1244 cm$^{-1}$ represent the C–O of the ether and ester, whereas the band at 3470 cm$^{-1}$ represents the elongation of the –OH group [21,41,50,51].

The esterification, epoxidation, and oxirane ring opening reactions were also confirmed through the [1]H NMR spectra (Figure 4). In the FACO sample (Figure 4), peak (a) is ascribed to the bonding of the hydrogen of the alkene to carbons 9 and 10 [7,52], whereas peak (b) is associated with the hydrogen of the terminal chain ($-CH_3$). Finally, peaks (c) and (d) correspond to the hydrogen bonded to the sp$^3$ carbon ($-CH_2-$) [7,53]. For the ERIC sample (Figure 3), the peak (e) corresponds to the hydrogen bonded to the carbon near the sp$^3$ oxygen of the ester functional group, confirming successful esterification. In the spectrum of the EPOXIRIC sample (Figure 3), peak (f), representing unsaturated bonds (still seen in the ERIC sample), has disappeared whereas peak (g) appears, indicating the formation of the epoxide ring [28,54]. Lastly, the spectra of OR2EHRIC and ORWATER samples (Figure 3) show the absence of peak (g) and the presence of peak (h), corresponding to the hydrogen bonded to the hydroxyl carbon ($-CH(OH)-$) [28,54].

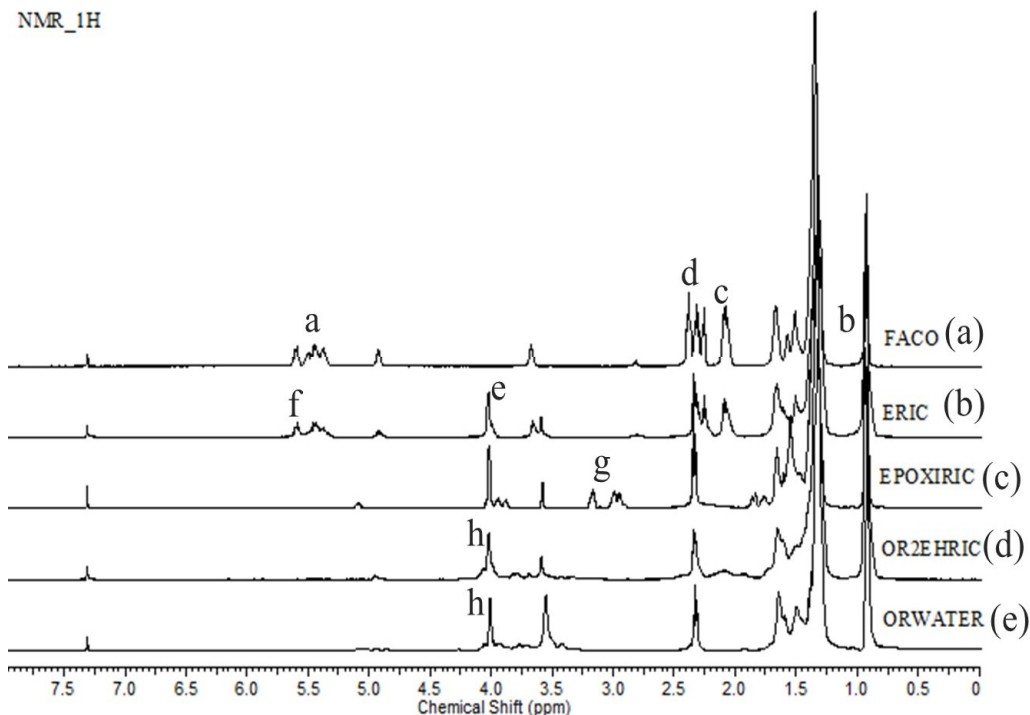

**Figure 4.** [1]H NMR spectra of the (a) FACO, (b) ERIC, (c) EPOXIRIC, (d) OR2EHRIC and (e) ORWATER samples.

### 3.2. Tribological Results

The FC curves and WSD values of the FACO, ERIC, OR2EHRIC, and ORWATER samples are shown in Figure 5, along with the results obtained for the mineral oil (SAE 20W50). The FC curves of the ORWATER and the mineral oil samples show essentially constant friction values. On the other hand, the FC curves of the FACO and OR2EHRIC samples gradually decrease with time. Interestingly, the FC curve of the ERIC sample fluctuates throughout the test. According to the concept of friction traces, steady state friction coefficients are classified into four types: type A—the FC is constant during the steady state; type B—the FC gradually increases during the test; type C—the FC gradually decreases during the test; and type D—the FC fluctuates throughout the test [55–57]. Type A is usually associated with low wear; for this reason, the ORWATER and the mineral oil samples show the lowest WSD values observed in this study, 281.2 ± 5.54 and 225.2 ± 2.16 μm, respectively [55,57]. Types B and C may be acceptable for lubricants, whereas type D is unacceptable because it is normally associated with high wear. As seen, the FACO and OR2EHRIC samples show friction curves with type C characteristics and thus exhibit intermediate WSD values. Finally, the ERIC sample presents a type D friction curve and thus exhibits the highest WSD observed in this study.

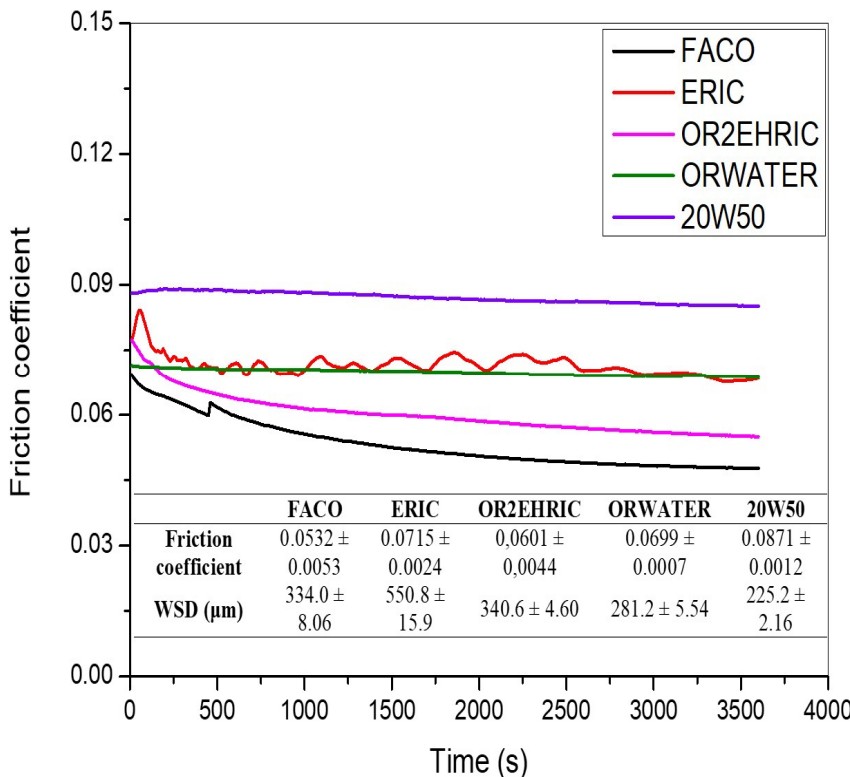

**Figure 5.** Friction coefficient curves in tribological testes of the bio-based samples, and of the commercial mineral oil sample (SAE 20W50).

All bio-based samples evaluated in this study present FC values lower than that of the 20W50 mineral oil sample. This may be attributed to the molecular structure of the biolubricants samples, which have polar and non-polar regions, whereas the 20W50 mineral oil contains predominantly non-polar molecules. The polar region of biolubricants is responsible for the absorption or adhesion to the metallic surfaces, and the non-polar region is responsible for the wear resistance of the lubricant film [13,58]. Previous studies suggest that lubricity and wear protection are affected by criteria related to molecular structure: degree of molecular branching, chain length, and number of ester functional groups [13].

In this study, the branching of the ERIC structure with 2-ethylhexanol (forming the OR2EHRIC structure) reduced the FC and WSD, corroborating those previously reported results. Additionally, the use of water as the nucleophilic agent to open the oxirane ring produced the most efficient biolubricant sample for wear reduction. The presence of three hydroxyls (–OH) in the ORWATER molecule improved the wear resistance of the biolubricant due to the strong molecular interactions induced by hydrogen bonding [10,13,59]. The WSD of the ORWATER sample (281.2 ± 5.54 μm) was close to that of the mineral oil (225.2 ± 2.16 μm), demonstrating its promising potential for use as a lubricant, especially considering that the commercial mineral oil sample (20W50) contains anti-wear additives that significantly improve the wear reduction capacity of mineral oil [60–62].

The morphologies of the wear surfaces of the studied samples are presented in Table 3. The grooves in the direction of application are caused by wear through abrasion and adhesion [26,63,64]. Among the bio-based samples, ORWATER showed the highest wear resistance of the lubricant film, demonstrating its stronger absorption on the metallic surfaces of the balls. Thus, the wear surfaces are smoother when using ORWATER. Although the 20W50 sample yielded the lowest WSD values, its wear morphology is visually rougher than those of the bio-based lubricants samples, which appear to be more efficient in forming a monomolecular (or multimolecular) layered structure oriented

toward the polar end, forming a film that inhibits the metal-to-metal contact. This would yield lower roughness of the worn surfaces [26,65,66].

**Table 3.** Wear morphology of the balls lubricated with the bio-based samples, and of the balls lubricated with the commercial mineral oil sample (SAE 20W50), after tribological tests.

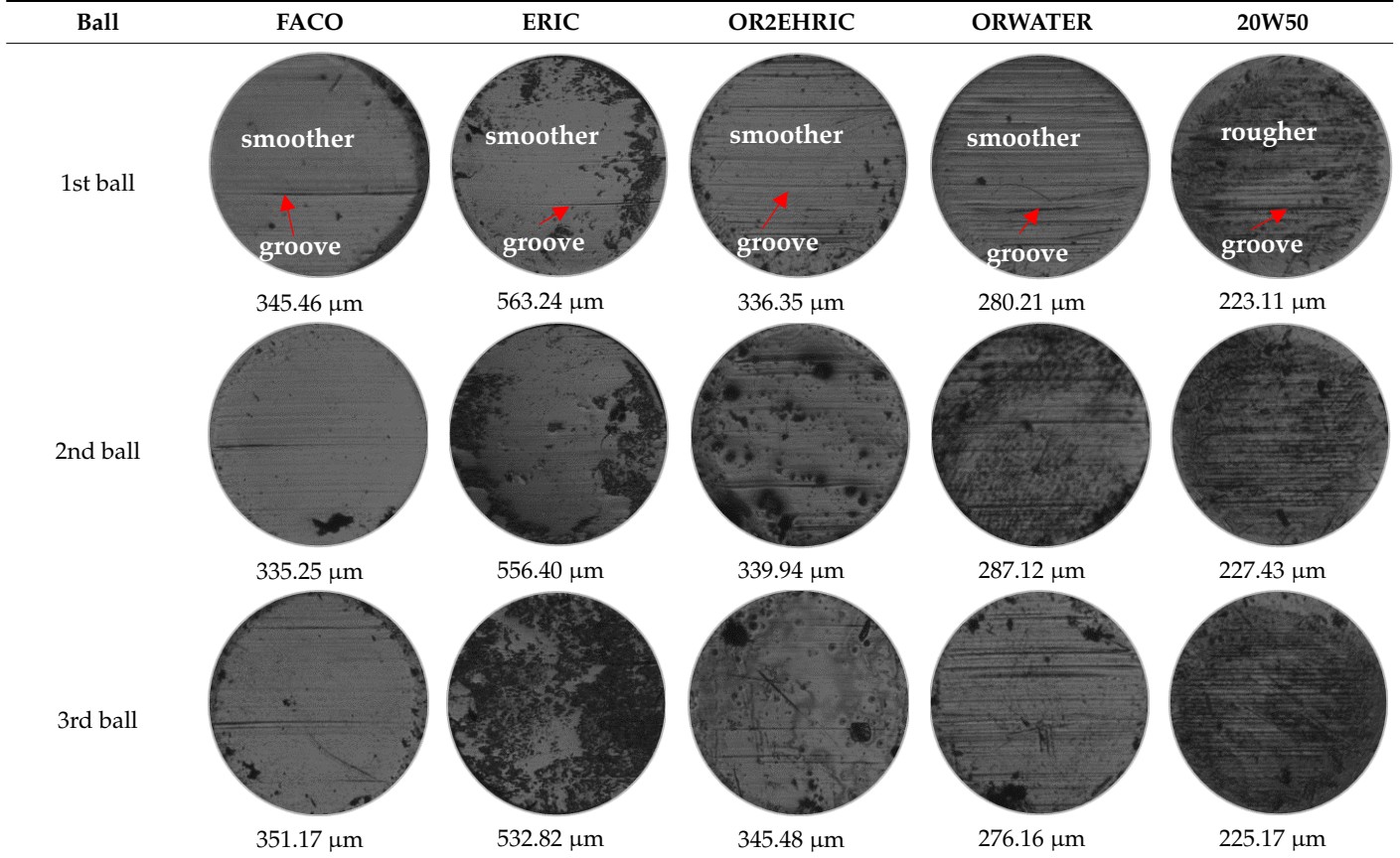

| Ball | FACO | ERIC | OR2EHRIC | ORWATER | 20W50 |
|---|---|---|---|---|---|
| 1st ball | smoother / groove / 345.46 µm | smoother / groove / 563.24 µm | smoother / groove / 336.35 µm | smoother / groove / 280.21 µm | rougher / groove / 223.11 µm |
| 2nd ball | 335.25 µm | 556.40 µm | 339.94 µm | 287.12 µm | 227.43 µm |
| 3rd ball | 351.17 µm | 532.82 µm | 345.48 µm | 276.16 µm | 225.17 µm |

## 4. Conclusions

In the present study, biolubricants samples were prepared through esterification (ERIC), epoxidation (EPOXIRIC), and oxirane ring opening reactions, using 2-ethylhexanol (OR2EHRIC) and water (ORWATER) as nucleophilic agents to create ether branches, and compared to a commercial mineral oil sample (SAE 20W50). The viscosities of the samples varied from 25.2 to 420.46 cSt at 40 °C and from 4.55 to 24.39 cSt at 100 °C. Successful syntheses were confirmed using FTIR and NMR analyses. The ORWATER and 20W50 samples showed constant FC curves throughout the tribological test, suggesting that they produce films that effectively reduce wear. The FC curve of the OR2EHRIC sample showed type C friction behavior, whereas that of the ERIC sample showed type D behavior. The ORWATER sample obtained from fatty acids of castor oil showed effective lubricant film formation capacity, with similar wear resistance and FC lower than that of the commercial mineral oil.

**Author Contributions:** Conceptualization, P.R.C.F.R.F. and M.R.d.N.; methodology, P.R.C.F.R.F., M.R.d.N. and S.S.O.d.S.; validation, P.R.C.F.R.F., M.R.d.N. and S.S.O.d.S.; formal analysis, F.M.T.d.L., E.R.-C. and C.L.C.J.; investigation, P.R.C.F.R.F.; resources, F.M.T.d.L., E.R.-C., and C.L.C.J.; data curation, P.R.C.F.R.F.; writing—original draft preparation, P.R.C.F.R.F., F.M.T.d.L., E.R.-C. and C.L.C.J.; writing—review and editing, P.R.C.F.R.F., F.M.T.d.L., E.R.-C. and C.L.C.J.; supervision, F.M.T.d.L., E.R.-C. and C.L.C.J.; project administration, F.M.T.d.L., E.R.-C. and C.L.C.J.; funding acquisition, F.M.T.d.L., E.R.-C. and C.L.C.J. All authors have read and agreed to the published version of the manuscript.

**Funding:** We thank CNPq (Conselho Nacional de Pesquisa e Desenvolvimento Científico) Projects: 304950/2019-0 (Tavares de Luna, F.M.), 309046/2018-1 (Loureiro Cavalcante, C., Jr.) and FUNCAP (Fundação Cearense de Apoio ao Desenvolvimento Científico e Tecnológico) Project E1-0079-0004301 for financial support for this study. Ribeiro-Filho thanks the FAPEMA (Fundação de Amparo à Pesquisa e ao Desenvolvimento Científico e Tecnológico do Maranhão) Project 01131/18 (Campos Flexa Ribeiro Filho, P.R.) and Universidade Estadual do Maranhão (UEMA) for financial support. E.R.-C. thanks Junta de Andalucía, project P20-00375 and FEDER funds.

**Data Availability Statement:** Not applicable.

**Acknowledgments:** We thank CNPq (Conselho Nacional de Pesquisa e Desenvolvimento Científico) and FUNCAP (Fundação Cearense de Apoio ao Desenvolvimento Científico e Tecnológico) for financial support for this study. Ribeiro-Filho thanks the FAPEMA (Fundação de Amparo à Pesquisa e ao Desenvolvimento Científico e Tecnológico do Maranhão) and Universidade Estadual do Maranhão (UEMA) for financial support. E.R.-C. thanks Junta de Andalucía (Spain) project P20_00375 and FEDER funds.

**Conflicts of Interest:** The authors declare no conflict of interest.

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
