# Peer review of "Synthesis and Frictional Characteristics of Bio-Based Lubricants Obtained from Fatty Acids of Castor Oil"

_lubricants, doi:10.3390/lubricants11020057_

Round 1

Reviewer 1 Report

This study is interesting and useful in describing tribological properties of different bio-based lubricants, with some interesting mechanistic insights presented as well. I suggest the authors consider the points below to improve quality of the article.

-       In introduction, line 45-48, authors attribute good lubrication properties of Ricinoleic acid to the presence of -OH groups. I wonder how a single -OH group can contribute to lubrication from a mechanistic point of view. The authors should justify this point in the manuscript.

-        In section 2.4, tribological evaluation, authors should clarify wear assessment protocol more clearly. How were the balls handled between different samples? Did you use new balls for each test? If you are reusing balls how can you compare wear tracks?

-       Results and discussion section, line 181-190, the discussion of viscosity based on molecular structure is interesting but is described in imprecise terms. For example, the sentence “…contains a hydroxyl group endowing it with a high molecular strength due to the hydrogen bonding.” can be more precisely described as “…contains a hydroxyl group that enhances attractive intermolecular interactions due to hydrogen bonding.” This paragraph should be revised.

Author Response

Please, see attached document.

Reviewer 2 Report

The present study describes the production bio-based lubricants from castor oil via a three-step process (esterification, epoxidation, and ring-opening reactions). The products were characterized via NMR analysis and their physico-chemical and tribological properties were also determined according to standard methods. The paper is highly relevant as content and logically built. I suggest that some points are addressed:

1. The authors should briefly report maximum conversion/yield percentage values for each step. Moreover, I recommend also report the experimental results (values) for physico-chemical and tribological properties. It is more interesting to the readers.

2. Introduction/last paragraph: How is this system different to other reports to merit publication? Please, report. A variety of strategies has been reported in the literature to prepare several bio-based lubricants.

3. Figure 1. The authors should report full nomenclature in Figure legend for all these abbreviations. This is strongly important to the readers.

Author Response

Please, see the attached document

Reviewer 3 Report

The authors have done a remarkable job citing the biodegradble lubricants. Though researchers have already worked in detail on castor oil but the authors have tried to bring out a novelty.However, this novelty is slightly weaker than the works done as compared to other researchers. It is quite interesting to see that the fatty acids proved to be better than the 20W50 oil. However, i am curious to note the performance of vegetable oil derivatives performing great than a commercial product. What was the viscosity of 20W50?. What is the roughness of the surfaces of the balls after the tests and before the test? Can you identify the lubrication regime using the lamda value? How many times the tests were repeated for the tribo test? Why did the authors take 40deg Celcius? Any specific reason?Also the loading was only 55N. Why such small loads and low temperatures? The chemical composition of 20W50 is for high loads and high temperature. The anti-wear test is also done at 75degree Celsius as per the ASTM test conditions. Which test conditions did the authors follow? Even though the viscosity index of FACO was less than ERIC, EPOXIRIC, OR2EHRIC, ORWATER but the frictional properties were better for FACO? Why ? As with an increase in temperature it is the VI helps to retain the lubricating properties but in this case it seems VI was not related to reduction in wear. Please explain. The surface morphologies needs to be more clear and is preferred in greyscale. Please indicate the damages on the surfaces. 

Overall, the article is good but the testing needs to be improved and needs to be done as per ASTM standards in order to compare with an industrial lubricant like the 20W50.

Author Response

Please, see the attached document

Round 2

Reviewer 2 Report

The authors corrected the manuscript as suggested. Therefore, I recommend its acceptance in present version.

Reviewer 3 Report

The authors have answered all the queries.